# Submicroscopic malaria in pregnancy and associated adverse pregnancy events: A case-cohort study of 4,352 women on the Thailand–Myanmar border

**Mary Ellen Gilder**[1,2]*, **Makoto Saito**[2,3], **Warat Haohankhunnatham**[1], **Clare L. Ling**[1,2], **Gornpan Gornsawun**[1], **Germana Bancone**[1,2], **Cindy S. Chu**[1,2], **Peter R. Christensen**[1], **Mallika Imwong**[4,5], **Prakaykaew Charunwatthana**[5,6], **Nay Win Tun**[1], **Aung Myat Min**[1], **Verena I. Carrara**[1,7], **Stephane Proux**[1], **Nicholas J. White**[2,5], **François Nosten**[1,2], **Rose McGready**[1,2]

1 Shoklo Malaria Research Unit, Mahidol-Oxford Tropical Medicine Research Unit, Faculty of Tropical Medicine, Mahidol University, Mae Sot, Thailand, 2 Centre for Tropical Medicine and Global Health, Nuffield Department of Medicine, University of Oxford, Oxford, United Kingdom, 3 Infectious Diseases Data Observatory (IDDO)/WorldWide Antimalarial Resistance Network (WWARN), Oxford, United Kingdom, 4 Department of Molecular Tropical Medicine and Genetics, Faculty of Tropical Medicine, Mahidol University, Bangkok, Thailand, 5 Mahidol-Oxford Tropical Medicine Research Unit (MORU), Faculty of Tropical Medicine, Mahidol University, Bangkok, Thailand, 6 Department of Clinical Tropical Medicine, Faculty of Tropical Medicine, Mahidol University, Bangkok, Thailand, 7 Institute of Global Health, Faculty of Medicine, University of Geneva, Geneva, Switzerland

* mellie@shoklo-unit.com

## Abstract

### Background

Malaria in pregnancy detected by microscopy is associated with maternal anaemia, reduced fetal growth, and preterm birth, but the effects of lower density (i.e., submicroscopic) malaria infections are poorly characterised. This analysis was undertaken to investigate associations between submicroscopic malaria at the first antenatal care (ANC) visit and these adverse pregnancy events on the Thailand–Myanmar border.

### Methods

Blood samples taken from refugee and migrant pregnant women presenting for their first ANC visit were analysed retrospectively for malaria using ultrasensitive PCR (uPCR, limit of detection 22 parasites/mL). The relationships between submicroscopic malaria and subsequent microscopically detectable malaria, anaemia, birth weight, and preterm birth were evaluated using inverse probability weighting for stratified random sampling.

### Results

First ANC visit samples from 4,352 asymptomatic women (median gestational age 16.5 weeks) attending between October 1st 2012 and December 31st 2015 were analysed. The weighted proportion of women with submicroscopic malaria infection was 4.6% (95% CI

**Data availability statement:** Data cannot be shared publicly because of the sensitivity of data for this population of undocumented refugees and migrants. De-identified participant data are available from the Mahidol Oxford Tropical Medicine Data Access Committee upon request from this link: https://www.tropmedres.ac/units/moru-bangkok/bioethics-engagement/datasharing.

**Funding:** The Shoklo Malaria Research Unit and this work was funded in part by the Wellcome-Trust Major Overseas Programme in Southeast Asia [220211]. MEG's DPhil is supported by the Tropical Network Fund of the University of Oxford. The funders had no role in study design, data collection and analysis, decision to publish, or preparation of the manuscript.

**Competing interests:** The authors have declared that no competing interests exist.

**Abbreviations:** ANC, antenatal care; CI, confidence interval; DNA, deoxyribonucleic acid; EGA, estimated gestational age; g, gram; HR, hazard ratio; IQR, interquartile range; mg, milligram; mL, millilitre; μL microlitre; mMiP, microscopy detected malaria in pregnancy; p, parasites; P., Plasmodium; pRBC, packed red blood cells; Q, quartile; sMiP, submicroscopic malaria in pregnancy; SMRU, Shoklo Malaria Research Unit; uPCR, ultrasensitive quantitative polymerase chain reactionRDTrapid diagnostic test

3.9–5.6), comprising 59.8% (49.5–69.4) *Plasmodium vivax,* 6.5% (4.0–10.5) *Plasmodium falciparum,* 1.8% (0.9–3.6) mixed, and 31.9% (22.2–43.5) infections which could not be speciated. Submicroscopic parasitaemia at first ANC visit was associated with subsequent microscopically detected malaria (adjusted hazard ratio [HR] 12.9, 95% CI 8.8–18.8, $p < 0.001$) and lower birth weight (adjusted predicted mean difference –275 g, 95% CI –510 to –40, $p = 0.022$). There was no association with preterm birth. Submicroscopic *P. falciparum* mono-infection (adjusted HR 2.8, 95% CI 1.2–6.6, $p = 0.023$) and coinfection with *P. falciparum and P. vivax* (adjusted HR 10.3, 95% CI 2.6–40.4, $p = 0.001$) was associated with increased risk of maternal anaemia, but submicroscopic *P. vivax* mono-infection was not. That uPCR was conducted for only a part of the cohort due to cost constraints is a limitation.

## Conclusions

In low transmission settings, uPCR identifies substantially more malaria infections at antenatal screening than conventional diagnostic methods. On the Thailand–Myanmar border, submicroscopic malaria at first antenatal consultation was associated with higher risks of microscopically diagnosed malaria later in pregnancy, anaemia, and reduced birth weight.

---

## Author summary
### Why was this study done?

- We know that malaria in pregnancy impacts mothers and their developing babies—especially by causing anaemia (low blood counts) and slowing the growth of the developing baby.

- Newer lab methods that detect malaria DNA (deoxyribonucleic acid) show that many people whose routine malaria tests at a clinic or hospital are negative actually have low-level infections in the blood.

- This study was done to see if these low-level infections have an impact on mothers and their unborn babies.

### What did the researchers do and find?

- The researchers used a very sensitive DNA amplification method that could detect even one parasite per drop of blood to see how many pregnant women were carrying parasites at their first antenatal visit.

- We found that about 1 in 20 pregnant women on the Thailand–Myanmar border had detectable malaria parasites, but 80% of malaria infections were very low-level infections missed by clinical tests.

- Pregnant women with these low-level malaria infections were more likely to get sick with malaria later in pregnancy, have anaemia, and give birth to small babies.

### What do these findings mean?

- These findings mean that most malaria infections are never detected or treated by healthcare workers, and the true impact of malaria on pregnant mothers and their developing babies may be much greater than previously understood.

- This sensitive DNA test is too complex and expensive to make it routinely available.

- Eliminating malaria from regions, countries, and—ultimately—the world may be the only way to truly eliminate the harmful effects of malaria in pregnancy.

## Background

Malaria infection with any species in pregnancy is harmful to both mother and fetus. Malaria is associated with poor pregnancy outcomes and with maternal and neonatal mortality [1–3]. Recurrent or symptomatic malaria or high parasite density infections have a greater adverse impact than single or asymptomatic infections [1,3]. However, even in low transmission settings, the majority of pregnancy malaria infections are asymptomatic. These low-density infections have also been linked to adverse outcomes including anaemia, poor fetal growth, and pregnancy loss although the evidence is less clear [1,3].

The use of polymerase chain reaction (PCR) methods to detect malaria has shown that up to 75% of PCR-detected malaria infections in low or unstable transmission settings are at parasite densities below the limit of detection by conventional routine light microscopy (i.e., submicroscopic malaria) [4–7]. The sensitivity of the different PCR methods used to detect malaria varies by orders of magnitude and depends on the volume of blood tested, the matrix (whole blood, packed red blood cells (pRBC), dried blood spots) and the method [8]. High volume, ultrasensitive quantitative PCR (uPCR) using 200 microlitres (μL) of pRBC can detect parasite densities as low as 22 parasites/millilitre (p/mL) whole blood [9], while the sensitivity of some PCR methods using dried blood spots are only slightly better than expert microscopy (50,000 p/mL whole blood) [8].

Since its introduction, PCR has been used increasingly in studies of epidemiology and malaria in pregnancy. Most of these investigations are from Africa where *Plasmodium falciparum* predominates [6]]. Relatively few studies address submicroscopic malaria in pregnancy (sMiP) in Asia and South America [10–14] where transmission is generally much lower and *Plasmodium vivax* predominates. There has been marked variation between studies in PCR methodology, limit of detection, and reporting of speciation, making comparisons between reports difficult. Some studies have demonstrated associations between sMiP and increased risk of maternal anaemia, lower infant birth weight, or preterm birth [10,15,16], whereas others have not [16–20].

Anaemia, preterm birth, and poor fetal growth leading to lower birth weights are all major public health problems in malaria endemic areas. These outcomes are all associated with lifelong negative impacts on offspring neurodevelopment and health [21–23]. The objective of this study was to test if these adverse pregnancy outcomes that are associated with microscopically detected malaria infections are also associated with submicroscopic malaria detected at the first antenatal care (ANC) visit. Here, we report the clinical and pregnancy outcomes from a well-characterised cohort of over 4,000 pregnant women from the Thailand–Myanmar border who were assessed for subclinical malaria by high-volume uPCR at their first ANC visit.

## Methods

### Setting

The study participants were migrant or refugee women attending the free ANC clinics of the Shoklo Malaria Research Unit (SMRU), which has operated on the border between Thailand and Myanmar since 1986. Malaria in pregnancy has been a significant problem locally, complicated by the presence of multidrug resistant strains of *P. falciparum*. This precludes chemoprophylaxis as there are no effective and proven safe drugs, and it limits therapeutic options

to some artemisinin-based combination therapies. *P. falciparum* was targeted for malaria elimination in this area and there have been no locally transmitted microscopic falciparum malaria infections in pregnant women since 2016. Chloroquine is the first line treatment for *P. vivax* infections, for which radical cure with primaquine has increased gradually in the general population as G6PD testing at field clinics became more available after 2017 [24]. Malaria has declined markedly in the region over the past 30 years. The rapidly changing epidemiology of malaria from 1995 to 2016 in non-pregnant patients seen at these border clinics has been reviewed previously [25].

## SMRU antenatal clinic

Blood smears to detect malaria at ANC visits were offered every 2 weeks and treatment was provided to microscopically diagnosed pregnant women with malaria regardless of symptoms. This was a retrospective evaluation of uPCR in initial blood samples which were processed after the women had delivered and were no longer in active care at SMRU. Anaemia screening was done regularly throughout the pregnancy (a minimum of three haematocrit measurements done at 2-week intervals initially, followed by haematocrits taken at 22, 28, and 36 weeks estimated gestational age (EGA)). Haematinic prophylaxis with ferrous sulphate (200 mg (mg) daily) and folic acid (5 mg once weekly) was routinely provided. Women with anaemia (haematocrit < 30%) received ferrous sulphate (400 mg twice per day) and folic acid (5 mg daily) for 12 weeks. Ultrasound was offered to all pregnant women routinely for determining EGA at first ANC visit. The women were encouraged to deliver with skilled birth attendants in the SMRU facilities where 24-h services including labour support were provided free of charge.

## Biobanking

During the study period (2012–2015), all women attending ANC for the first time were screened for malaria by microscopy and offered screening bloodwork (including screening for HIV, hepatitis B, syphilis, and anaemia). Counselling regarding biobanking leftover samples for additional investigations was provided, and verbal consent was obtained. Use of the biobank for malaria uPCR was approved by both local and international ethics boards (see below). The only exclusion criterion for this biobank was if there was a severe medical or obstetric emergency at presentation. Blood samples from this first ANC biobank were extracted separately from three groups:

Group 1: from pregnant women who had microscopically detected malaria in pregnancy (mMiP) detected at any time after the first ANC visit

Group 2: pregnant women with anaemia, but no mMiP

Group 3: non-case pregnant women who did not have mMiP or anaemia detected at any time during the study period.

Women with mMiP at or before their first ANC visit were excluded from the analysis.

## Study design

All samples from group 1 (mMiP detected after the first ANC visit); and selected samples from the much larger groups 2 and 3 (anaemia detected after first ANC visit, and non-case women) were analysed. Samples from groups 2 and 3 were selected by random extraction from purposively selected time blocks in order to accommodate trends in submicroscopic parasitaemia over time. To assess the changes over time, approximately 500 samples were

randomly selected from eight different time blocks: six consecutive quarters between October 2013 and March 2015, and the 4th quarters (i.e., October–December) of 2012 and 2015.

Samples were noted as unavailable if not stored in the biobank, e.g., if incorrectly labelled and not identified as a first ANC sample, or if they were used for other urgent clinical investigations at the time of the first ANC consultation.

This study did not have a formal analysis plan. The preliminary analysis was planned as a cohort study. However, considering that uPCR was conducted only in a subset of the whole cohort using stratified random sampling due to cost constraints, the data was analysed as a case-cohort study using inverse sampling-probability weighting as described below based on early input from co-authors. All exposures and outcomes were pre-defined prior to analysis, while specific cut-offs for analysis were defined by the data where appropriate (e.g., parasitaemia quartiles).

This study is reported as per the Strengthening the Reporting of Observational Studies in Epidemiology (STROBE) guideline (S1 STROBE Checklist).

### Ethics approval and consent to participate

Retrospective analysis of ANC data and biobanked samples was approved by the Ethics Committee of the Faculty of Tropical Medicine at Mahidol University (Ethics reference: TMEC 17–027) and Oxford Tropical Research Ethics Committee (Ethics reference: OxTREC 583-16). Participants freely chose to access ANC care at SMRU clinics and were informed at their first ANC visit that they had the right to opt out of all screening procedures.

### Laboratory methods

Microscopy with Giemsa stain was used for detecting parasitaemia at each visit. Malaria parasites counted per 1,000 red blood cells (thin smear) or per 500 white blood cells (thick smear). Negative smears were declared after 200 high power fields were read on the thick blood smear. This method has a limit of detection of 10–50 p/μL (10,000–50,000 p/mL) [26]. Laboratory technicians had regular quality control as part of ongoing studies during this period.

A 3-mL venous blood sample was collected from each participant in an EDTA Vacutainer. After mixing, venous samples were stored and transported at approximately 4 °C to the central SMRU laboratory in Mae Sot, Thailand, within 48 h. After routine complete blood count and haemoglobin (Hb) typing, the remaining whole blood was centrifuged and an aliquot of pRBC was stored at −80 °C. Ultrasensitive qPCR detection was done on DNA extracted from 200 μL pRBC (equivalent to ∼600 μL whole blood at a haematocrit of 33%) using QIAamp blood minikit (Qiagen, Germany) as previously described [9]. Standard curves using controls of known density of malaria ring-stage-infected red blood cells were used to calibrate each PCR run, allowing parasite density to be calculated from PCR cycle thresholds. The lower limit of accurate quantitation of this method is 22 p/mL of whole blood. Malaria species was determined by nested species-specific PCR where possible [9]. If the parasite density was low, speciation was often not possible.

Haematocrit was obtained from finger prick blood samples and read with a Hawksley scale on a capillary tube sample centrifuged at 10,000 rotations per minute for 3 min. Haemoglobin typing was determined on EDTA blood either by Capillary Electrophoresis (using a Capillarys II, Sebia, France) at the central SMRU Haematology Laboratory or at an external laboratory.

### Definition of outcomes and inclusion criteria for each outcome

We analysed four outcomes: mMiP, anaemia, birth weight, and preterm birth. mMiP was defined as microscopically detected malaria of any species by active screening as described

above regardless of the previous uPCR result. Anaemia was defined as a haematocrit value of <30% at any point in the pregnancy after the first ANC visit. Women with anaemia at first ANC were excluded from the analysis on anaemia.

Women were included in the birth weight analysis if their infants were live-born singletons without congenital abnormality, had a dating ultrasound before 24-weeks EGA, and if they had an infant weight measured within 72 h of birth. Standard practice at SMRU clinics was to weigh the infant within 1 h of birth. A supplemental analysis was conducted using z-scores of birth weight accounting for sex and gestational age based on the INTERGROWTH-21st standards [27].

Preterm birth was defined as delivery before 37 weeks gestational age. Women were included in the analysis of preterm birth if they had a dating ultrasound before 24 weeks and their infants were live-born singletons without congenital abnormality. Women who were lost to follow up before delivery were censored at the day they were lost.

### Exposure

An uPCR result with parasite density of ≥22 p/mL was considered positive. If a malaria smear on the same day was negative by microscopy, this was considered sMiP. The primary analysis differentiated submicroscopic malaria species except when specified.

Submicroscopic infections were divided into four parasite density strata (Q) for subgroup analysis—Q1: 22–106 p/mL, Q2: 107–367 p/mL; Q3: 368–2,003 p/mL; Q4: >2,003 p/mL. When data were sparse, infections were grouped as being above or below the median value 367 p/mL.

### Key covariate: Haemoglobin typing

Data from Hb typing was categorised to create an ordinal variable as follows: normal or mild (normal or alpha thalassaemia trait), moderate (beta thalassaemia trait and Hb E trait), and severe (Hb EE homozygotes, HbH, beta thalassaemia with HbE disease). Moderate and severe categories were combined when data were sparse.

For detailed description of data extraction and other covariates, see the Supplementary methods (S1 Text).

### Statistical methods

Time-fixed inverse probability weighting was used to account for differential stratified sampling probability for each sub-cohort—microscopically diagnosed malaria, anaemia, and non-case [28]. Weights equal to the reciprocal of the sampling fraction for each sub-cohort were calculated separately for each 3–6-month block, and for migrants and refugees to account for differing malaria transmission intensities over time and by location. Maternal characteristics of the sub-cohort and the prevalence of sMiP at the first ANC over time were described by weighted frequencies (proportion with 95% confidence interval) or weighted median with interquartile range [IQR]. Weighted characteristics of the sampled women were compared with characteristics of the cohort as a whole to assess balance. Balance was additionally checked by comparing characteristics of sampled versus unsampled participants. As exploratory analyses, associations between the primary exposure (sMiP at first ANC) and other baseline characteristics were assessed by univariable and multivariable inverse-probability-weighted logistic regression.

Causal models for the relationship of sMiP with mMiP, anaemia, birth weight, and preterm birth were built using directed acyclic graphs. Covariates for regression models were identified through analysis of these causal models.

The risk of developing mMiP, anaemia or preterm birth based on the presence or absence of sMiP at first ANC visit was calculated using inverse-probability-weighted Cox regression. The sandwich estimator was used for robust standard error.

Women were censored at delivery, miscarriage, or at last visit seen before loss to follow up. Survival analysis started from the EGA (in days) of the first ANC visit. For analysis of anaemia-free survival among women with sMiP, women who developed mMiP were censored on the day before their first mMiP episode. For the analysis of anaemia and mMiP, survival analysis was terminated at 210 days (30 weeks) of follow up as most women had delivered by then, resulting in few data beyond this point. For analysis of preterm birth, survival analysis was terminated at EGA of 37 + 0 weeks (259 days). The proportional hazard assumption was assessed by Schoenfeld residuals, and if violated, a stratified Cox analysis was conducted.

Analysis of the association between sMiP and birth weight was done using weighted linear regression, with the delta method used to calculate the linearised standard error for the weighted data.

Because most variables had missing data for <15 individuals (0.1%) (except for haemoglobin variants, which had data missing for 292 women (2.4%)) complete case analysis was conducted.

Stata IC 15 [29] was used for statistical analyses

## Results

### Description of the cohort

The SMRU ANC clinics registered and screened 12,034 women from October 1, 2012 to December 31, 2015. From the collected biobank 4,352 stratified samples were selected for malaria uPCR (Fig 1). Maternal characteristics and birth outcomes in the full cohort were well represented in the subgroups of women selected for uPCR (S2 Text and S1 Table).

### Baseline characteristics at the first antenatal care visit in the sub-cohort

The median weighted age (IQR) at first ANC was 26 (21–31) years and EGA was 16.5 (9.9–25.6) weeks. Anaemia (weighted proportion 13.5%), smoking (13.3%), and low BMI (10.6%) were common. One-third of women were primiparous (30.9%), and 57.7% of the women reported that they were literate. Women were followed for a weighted median of 116 (IQR 36–183) days. Approximately one in five women were lost to follow up before delivery. This was less common among women with pregnancies complicated by anaemia or malaria (Table 1).

### Submicroscopic malaria infection at the first antenatal care visit

Overall, the weighted proportion of women with sMiP at first ANC visit in the cohort was 4.6% (95% CI 3.9–5.6), compared with microscopically detected malaria which was found in 1.1% (131/12,034) of women presenting for their first ANC. Within each subgroup, the prevalence of submicroscopic malaria differed: 36.1% (65/180) of women in group 1 (women who had subsequent microscopically detected malaria during this pregnancy) compared with 5.1% (30/592) in group 2 (anaemic women) and 3.8% (136/3,580) in group 3 (non-case women: i.e., those in whom neither mMiP nor anaemia was detected during their pregnancy). Thus, the odds ratios (95% CI) for subsequent patent malaria and anaemia were 14.31 (10.11–20.27) and 1.35 (0.90–2.02) compared with non-cases.

Among women with sMiP at first ANC, weighted proportions (95% CI) of each species were *P. vivax* 59.8% (49.5–69.4), *P. falciparum* 6.5% (4.0–10.5), mixed infections with both *P. falciparum* and *P. vivax* 1.8% (0.9–3.6), and unspeciated 31.9% (22.2–43.5). Proportions of

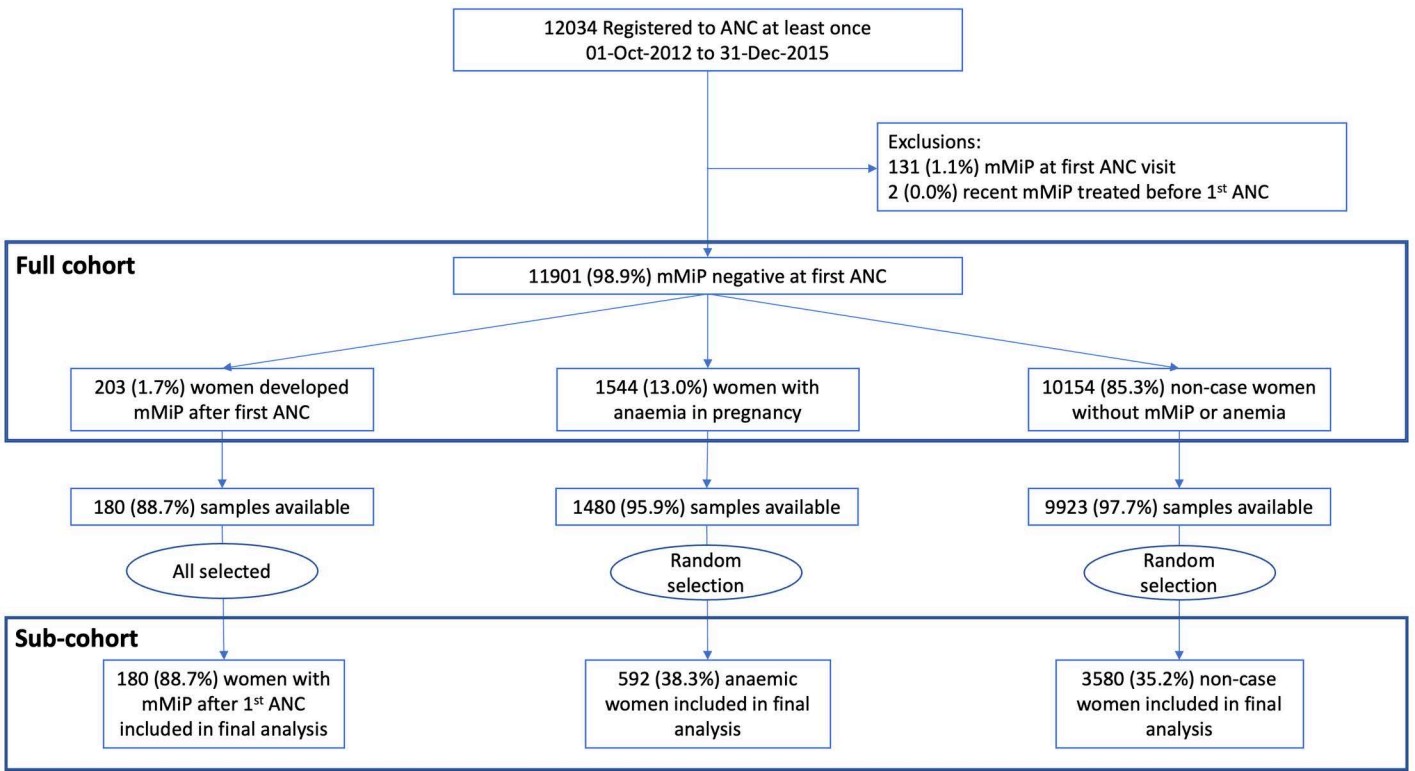

**Fig 1. Study flow.** ANC, antenatal care; mMiP, microscopic malaria in pregnancy.

species for microscopically detected infections at first ANC visit were 64.9% (85/131) *P. vivax*, 31.3% (41/131) *P. falciparum*, and 3.8% (5/131) were mixed infections with both species.

Parasite density estimates among women with submicroscopic infections ranged from 22 to 186,048 p/mL with a geometric mean of 515 p/mL. The majority of the infections fell below the limit of detection for all conventional methods, including PCR from dried blood spots (Fig 2). Dried blood spot based PCR methods with a limit of detection of 5,000 p/mL would miss 84% (195/231) of these infections, while improving the limit of detection to 1,000 p/mL would still miss 66% (152/231).

## Risk factors for submicroscopic malaria

Both microscopically detected and submicroscopic infections were more common in migrant women than in refugees (who lived in areas where malaria elimination efforts were more advanced). This association remained after adjusting for confounders (S2 Table, sMiP only). However, among the smaller number of women with mMiP in the refugee camp almost two-thirds (63.3% (19/30)) had preceding sMiP detected at first ANC visit. In comparison, less than a third of migrant women (living in communities with higher malaria transmission) who developed mMiP had sMiP at their first ANC visit (30.7% (46/150), $p = 0.001$).

Primigravid women (compared with those who had four or more pregnancies), women enrolled in earlier years of the cohort, and smokers were more likely to have a submicroscopic infection at their first ANC visit. Though lower gravidity was associated with higher prevalence of sMiP, both primigravidae and multigravidae were affected, and women with 2–3 pregnancies had an equal prevalence of sMiP compared with primigravidae (S1 Fig). Both

**Table 1. Cohort characteristics.**

| Demographics | | Non-cases ($n$ = 3,580) | mMiP ($n$ = 180) | Anaemia ($n$ = 592) |
|---|---|---|---|---|
| Age group | <20 | 590 (16.5) | 56 (31.1) | 93 (15.7) |
| | 20–29 | 1,857 (51.9) | 71 (39.4) | 271 (45.8) |
| | ≥30 | 1,133 (31.7) | 53 (29.4) | 228 (38.5) |
| Gravidity group | 1 | 1,137 (31.8) | 74 (41.1) | 155 (26.2) |
| | 2–3 | 1,439 (40.2) | 52 (28.9) | 194 (32.8) |
| | ≥4 | 1,004 (28.0) | 54 (30.0) | 243 (41.1) |
| Literacy | | 2,197 (61.4) | 91 (50.6) | 316 (53.4) |
| Smoking | | 469 (13.1) | 32 (17.8) | 103 (17.4) |
| Migrant | | 1,929 (53.9) | 150 (83.3) | 247 (41.7) |
| Low body mass index (<18.5 kg/m²) | | 354 (9.9) | 19 (10.6) | 70 (11.8) |
| Body mass index (kg/m²), median [IQR] | | 22.2 [20.1–24.5] | 21.3 [20.1–23.0] | 21.4 [19.6–23.3] |
| First ANC in trimester 1 | | 1,464 (40.9) | 90 (50.0) | 253 (42.7) |
| EGA 1st ANC, weeks median [IQR] | | 17 [10–26] | 13 [9–21] | 17 [10–26] |
| Microscopically detected malaria after 1st ANC | | 0 (0) | 180 (100) | 0 (0) |
| Submicroscopic malaria | | 136 (3.8) | 65 (36.1) | 30 (5.1) |
| *P. vivax* | | 101 (74.3) | 49 (75.4) | 11 (36.7) |
| *P. falciparum* | | 8 (5.9) | 3 (4.6) | 9 (30.0) |
| Unspeciated *Plasmodium* | | 26 (19.1) | 8 (12.3) | 8 (26.7) |
| Mixed infections | | 1 (0.7) | 35 (7.7) | 2 (6.7) |
| Geometric mean density (p/mL) [95% CI] | | 264 [201–347] | 1,597 [945–2,698] | 910 [411–2,016] |
| Pre-eclampsia/eclampsia | | 64 (1.8) | 4 (2.2) | 14 (2.4) |
| Anaemia in pregnancy | | 0 (0) | 58 (32.2) | 292 (100) |
| Haematocrit measures, median [IQR] | | 4 [2–7] | 11 [7–16] | 9 [5–13] |
| Haemoglobin variants[†] | Normal or mild | 2,945 (88.8) | 161 (91.5) | 372 (68.4) |
| | Moderate | 370 (11.2) | 15 (8.5) | 165 (30.3) |
| | Severe | 3 (0.1) | 0 (0) | 7 (1.3) |
| Delivery outcome (missing = 920, reported as lost to follow up) | Delivery | 2,458 (68.7) | 151 (83.9) | 480 (81.1) |
| | Twins | 32 (0.9) | 1 (0.6) | 8 (1.4) |
| | Miscarriage | 279 (7.8) | 6 (3.3) | 17 (2.9) |
| | Lost to follow up | 811 (22.7) | 22 (12.2) | 87 (14.7) |

Data are $n$ (%) unless otherwise specified. Data are complete except where specified as missing.

Missing data: BMI $n$ = 11 (2 anaemia, 9 non-case).

Malaria smear $n$ = 1 (uPCR negative, abortion case).

Hb variants $n$ = 292 (262 non-case, 4 mMiP, 48 anaemia).

[†]For details about groups, see the results section.

Abbreviations: EGA, estimated gestational age; ANC, antenatal care; IQR, interquartile range; mMiP, microscopically detected malaria in pregnancy.

microscopically detected (not shown) and submicroscopic infections with *P. vivax* were present year-round despite seasonality of rainfall. Overall trends are shown in the S2 and S3 Figs.

## Risk of subsequent microscopically detected malaria after submicroscopic infection

Patent mMiP after first ANC was seen more than 10 times more often in women with sMiP of any species than in those with a negative uPCR at the first ANC visit (13.2% (95% CI

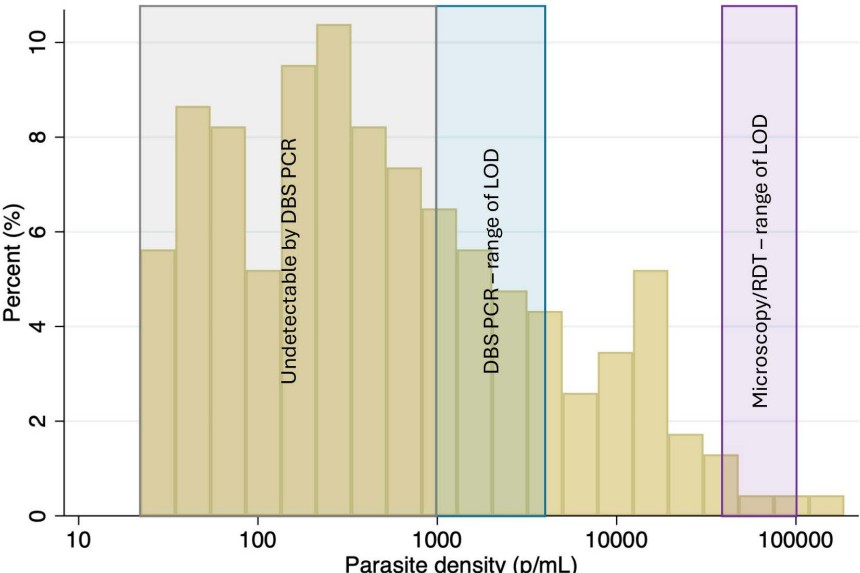

**Fig 2. Parasite densities among women with positive uPCR and negative microscopy.** This histogram shows the percent of the total positive tests vs. parasite density (on a logarithmic scale) demonstrating the high proportion that would be undetectable by PCR from dried blood spots (DBS) due, largely, to small sample volumes. LOD: limit of detection. RDT: rapid diagnostic test.

**Table 2. Submicroscopic malaria infection at first antenatal care visit and adjusted hazard ratio (HR) of developing patent microscopic malaria in pregnancy.**

| | | Weighted proportion with mMiP after first ANC (95% CI) | Unadjusted HR for mMiP (95% CI) | *p*-value | Adjusted HR for mMiP (95% CI)* | *p*-value |
|---|---|---|---|---|---|---|
| uPCR at first ANC | | | | | | |
| Negative | | 1.1 (1.0–1.4) | Reference | | Reference | |
| *P. vivax* | | 16.6 (12.3–22.0) | 19.3 (13.2–28.2) | <0.001 | 15.1 (9.8–23.5) | <0.001 |
| *P. species (not differentiable)* | | 5.1 (2.3–11.1) | 6.1 (2.5–15.0) | <0.001 | 5.8 (2.4–13.5) | <0.001 |
| *P. falciparum* | | 9.4 (2.9–26.4) | 8.7 (2.5–30.2) | 0.001 | 9.5 (2.8–32.7) | <0.001 |
| Mixed (*P. falciparum* and *P. vivax*) | | 57.3 (24.2–84.9) | 123.5 (59.9–254.6) | <0.001 | 67.2 (32.7–138.7) | <0.001 |
| Status | Refugee | 0.7 (0.5–1.1) | Reference | | – | |
| | Migrant | 2.3 (1.9–2.7) | 3.9 (2.6–5.9) | <0.001 | *Adjusted** | |
| Gravidity | Multigravida | 1.5 (1.2–1.8) | Reference | | Reference | |
| | Primigravida | 2.3 (1.8–2.9) | 1.7 (1.2–2.3) | 0.001 | *1.5 (1.1–2.2)* | 0.025 |
| Year of enrolment | 2012–2013 | 2.5 (2.1–3.1) | Reference | | Reference | |
| | 2014–2015 | 1.2 (1.0–1.5) | *0.5 (0.3–0.7)* | <0.001 | 0.4 (0.3–0.6) | <0.001 |
| Smoking | Non–smoker | 1.6 (1.4–1.9) | Reference | | Reference | |
| | Smoker | 2.3 (1.6–3.3) | *1.7 (1.1–2.5)* | 0.013 | 1.7 (1.1–2.7) | 0.018 |
| Literacy | Literate | 1.5 (1.2–1.8) | Reference | | Reference | |
| | Illiterate | 2.0 (1.6–2.5) | 1.3 (1.0–1.8) | 0.084 | 1.0 (0.7–1.4) | 0.956 |

*N* = 4,352. Exclusions: none. Survival follow up was terminated at 210 days because of sparse data after this point.

Abbreviations: ANC, antenatal care; mMiP, microscopically detected malaria in pregnancy; *P*, *Plasmodium*; uPCR, ultrasensitive quantitative polymerase chain reaction.

*Final model stratified for refugee/migrant status.

9.9–17.3) versus 1.1% (95% CI 1.0–1.4)). After controlling for confounders which included migrant or refugee status, gravidity, year of enrolment, smoking and literacy, the strong association remained for sMiP overall (adjusted HR 12.9, 95% C1 8.8–18.8, $p < 0.001$), and for each malaria species (Table 2). Submicroscopic infections including *P. vivax* were most likely to be followed by an episode of mMiP (adjusted HR for submicroscopic *P. vivax* 15.1, 95% CI 9.8–23.5, $p < 0.001$; and for mixed infection 67.2, 95% CI 32.7–138.7, $p < 0.001$) (S3 Table). Submicroscopic infection with any species increased the risk of subsequent microscopically diagnosed *P. vivax* infection: submicroscopic *P. vivax* (adjusted HR 17.4, 95% CI 11.0–27.5), unspeciated malaria (adjusted HR 6.1, 95% CI 2.5–14.7), *P. falciparum* (adjusted HR 7.4, 95% CI 1.7–32.4), and mixed infection (adjusted HR 16.4, 95% CI 2.0–132.4). This was not the same for *P. falciparum*. After adjusting for status, gravidity, year, smoking, and literacy, only submicroscopic infection with *P. falciparum* (adjusted HR 33.4, 95% CI 4.5–249.4) or mixed infections (adjusted HR 278.9, 95%C CI 73.3–1061.0) were significantly associated with subsequent microscopically diagnosed *P. falciparum*. The increased risk of mMiP of any species following submicroscopic *P. falciparum* was limited to the first 2 months of follow up, while the risk for *P. vivax* and mixed infections remained elevated throughout the pregnancy (Fig 3 and S4). Overall the majority of women with sMiP at first ANC did not have a subsequent episode of mMiP detected (weighted proportion 86.8%, 95% CI 82.7–90.1).

Among women with submicroscopic parasitaemia, higher parasite densities were associated with an increased risk of subsequent mMiP. This relationship was still evident after controlling for refugee status, gravidity, year, smoking, and literacy: Q1: 22–106 p/mL, adjusted HR 5.4 (95% CI 2.4–12.3); Q2: 107–367 p/mL, adjusted HR 7.0 (95% CI 3.3–14.7); Q3: 368–2,003 p/mL, adjusted HR 17.9 (95% CI 9.7–32.8); Q4: >2,003 p/mL, adjusted HR 27.7 (95% CI 15.5–49.4), *p*-values for all <0.001.

## Risk of anaemia in pregnancy with submicroscopic malaria infection

The risk of anaemia subsequent to the first ANC visit was similar in the combined group of women with sMiP (all species: weighted frequency 7.3%, 95% CI 4.3–12.4) and women with negative uPCR (9.7%, 95% CI 7.7–12.3, $p = 0.341$). However, in the smaller subgroup women presenting with submicroscopic *P. falciparum* infections there was a strong association with subsequent anaemia. The weighted proportion of women developing anaemia following detection of submicroscopic *P. falciparum* infection was 30.2% (95% CI 11.8–53.2), three times higher than the background incidence. Hazard ratios (95% CI) adjusted for parity, refugee status, literacy, year of enrolment, smoking, and haemoglobin variants were 2.8 (1.2–6.6) for *P. falciparum* monoinfection and 10.3 (2.6–40.4) for mixed infection (Table 3 and Figs 4 and S4). The median (IQR) haematocrit nadir for women with *P. falciparum* sMiP was 30% (29%–32%), compared with 32% (30%–35%) for women with no sMiP. This is equivalent to a haemoglobin reduction of approximately 0.6 g/dl (95% CI 0.2–1.1). There was no increase in the subsequent risk of anaemia in women with submicroscopic *P. vivax* or unspeciated malaria parasitaemias.

Submicroscopic infections with *P. falciparum* or mixed species that had parasite densities above the median (367 p/ml) increased the risk of anaemia (adjusted HR 9.3, 95% CI 2.9–30.0, $p < 0.001$) after adjusting for parity, refugee status, literacy, year of enrolment, smoking, and haemoglobin variants, while lower parasite densities did not (adjusted HR 1.9, 95% CI 0.7–5.2, $p = 0.236$). Neither higher (adjusted HR 1.7, 95% CI 0.9–3.2, $p = 0.123$) nor lower (adjusted HR 0.6, 95% CI 0.3–1.4, $p = 0.253$) submicroscopic parasite densities with *P. vivax* or unspeciated malaria were significantly associated with anaemia. The geometric mean parasite density at first ANC visit of women with sMiP who developed anaemia was 1,047 (95% CI 508–2,157) p/mL.

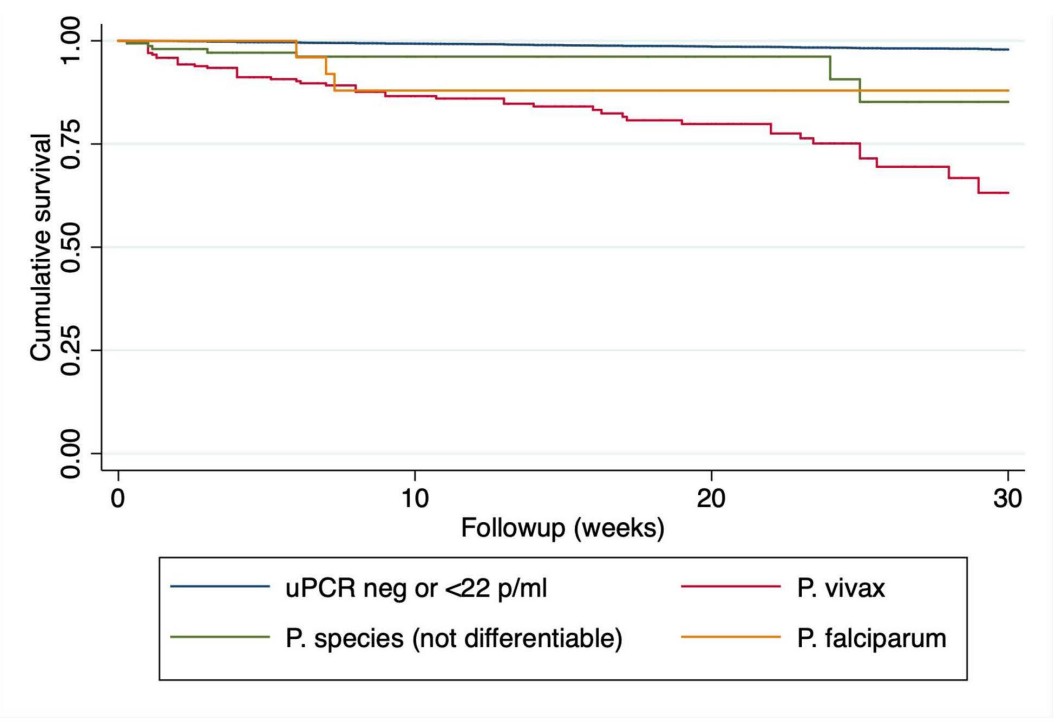

| | Follow-up interval (months) | | | | | | | |
|---|---|---|---|---|---|---|---|---|
| | 0-<1 | 1-<2 | 2-<3 | 3-<4 | 4-<5 | 5-<6 | 6-<7 | 7-<7.5 |
| uPCR neg at risk | 3577 | 2726 | 2324 | 1949 | 1559 | 1139 | 749 | 309 |
| uPCR neg mMiP | 19 | 24 | 16 | 21 | 11 | 10 | 7 | 4 |
| uPCR PV at risk | 153 | 105 | 83 | 66 | 47 | 36 | 21 | 6 |
| uPCR PV mMiP | 17 | 9 | 6 | 3 | 5 | 4 | 3 | 2 |
| uPCR P. spp at risk | 40 | 29 | 22 | 17 | 15 | 11 | 8 | 2 |
| uPCR P. spp mMiP | 4 | 1 | 0 | 0 | 0 | 0 | 2 | 0 |
| uPCR PF at risk | 19 | 14 | 11 | 11 | 8 | 6 | 4 | 3 |
| uPCR PF mMiP | 0 | 3 | 0 | 0 | 0 | 0 | 0 | 0 |

**Fig 3. Weighted survival without microscopically detected malaria following a positive uPCR result by uPCR species detected.** Survival terminated at 30 weeks of follow up because of sparse data beyond that time. Mixed infections omitted because of small numbers. (For results including mixed infections, see S4 Fig).

### Pregnancy outcomes: Birth weight and preterm birth

One thousand nine hundred ninety-six women met the eligibility criteria for analysis of birth weight. Lack of an early ultrasound to determine EGA (1,242 women) and loss to follow up (628 women) were the most common reasons for exclusion from this assessment. The weighted mean (95% CI) birth weight among term (EGA ≥ 37 weeks and 0 days) pregnancies was 3,081 g (3,053–3,108 g) for the 1,714 eligible uninfected women, 2,937 g (2,834–3,039 g) for the 52 eligible women with sMiP, and 2,971 g (2,896–3,046 g) for the 101 eligible women with mMiP (S5A Fig). When birth weight was corrected for gestational age and infant sex, mean

**Table 3. Submicroscopic malaria infection at first antenatal care visit and adjusted hazard ratio (HR) of developing anaemia in pregnancy.**

| | | Median HCT nadir of weighted data (IQR)[#] | Weighted proportion with anaemia[#] | Unadjusted HR for anaemia (95% CI) | *p*-value | adjusted HR for anaemia (95% CI), *p*-value | *p*-value |
|---|---|---|---|---|---|---|---|
| **uPCR category** | | | | | | | |
| Negative | | 32 (30–35) | 9.7 (7.7–12.3) | Reference | – | Reference | – |
| *P. vivax* | | 32 (30–34) | 4.8 (2.3–10.0) | 0.9 (0.5–1.5) | 0.608 | 0.9 (0.5–1.6) | 0.665 |
| *P. species (not differentiable)* | | 33 (31–34) | 7.8 (2.5–21.4) | 1.3 (0.4–3.7) | 0.658 | 1.4 (0.5–3.7) | 0.486 |
| *P. falciparum* | | 30 (29–32) | 30.2 (11.8–58.2) | 3.3 (1.3–8.5) | *0.012* | 2.8 (1.2–6.6) | *0.023* |
| Mixed (*P. falciparum* and *P. vivax*)[††] | | n.a. | n.a. | 7.3 (2.9–18.4) | <0.001 | 10.3 (2.6–40.4) | 0.001 |
| **Status** | Refugee | 32 (30–34) | 13.1 (11.0–15.7) | Reference | – | Reference | – |
| | Migrant | 33 (31–35) | 7.6 (4.9–11.5) | 0.7 (0.5–1.1) | 0.182 | *0.6 (0.4–0.9)* | *0.006* |
| **Gravidity** | G < 4 | 33 (31–35) | 7.3 (5.9–9.0) | Reference | – | Reference | – |
| | G ≥ 4 | 32 (30–35) | 15.3 (10.0–22.7) | *2.2 (1.4–3.6)* | *0.001* | 2.2 (1.4–3.5) | *0.001* |
| **Year of enrolment** | 2012–2013 | 32 (30–35) | 8.9 (6.5–12.1) | Reference | – | Reference | – |
| | 2014–2015 | 32 (30–35) | 10.0 (7.3–13.5) | 1.0 (0.7–1.6) | 0.848 | *1.0 (0.7–1.4)* | *0.836* |
| **Smoking** | Non–smoker | 32 (30–35) | 9.8 (7.6–12.6) | Reference | – | Reference | – |
| | Smoker | 32 (30–35) | 8.4 (6.0–11.7) | 1.1 (0.7–1.7) | 0.725 | 0.6 (0.4–1.0) | 0.059 |
| **Literacy** | Literate | 32 (31–35) | 7.2 (6.0–8.7) | Reference | – | – | – |
| | Illiterate | 32 (30–35) | 13.0 (8.9–18.5) | *1.9 (1.3–2.9)* | *0.002* | *Adjusted**[**] | – |
| **Hb Variants** | Normal/mild | 33 (31–35) | 7.0 (5.5–8.8) | Reference | – | Reference | – |
| | Moderate | 31 (29–34) | 25.8 (16.9–37.4) | *4.2 (2.5–7.0)* | *<0.001* | *4.5 (2.8–7.2)* | *<0.001* |
| | Severe† | 30 (29–30) | 34.2 (7.3–77.4) | | | | |

*N* = 3,676. Exclusions: anaemia at first ANC (*n* = 210), and data on Hb Variants missing (*n* = 292). Survival follow up was terminated at 210 days because of sparse data after this point.

[#]Women who developed mMiP excluded.

[††]n.a. not available due to sparse data (5/8 mixed infections were excluded due to subsequent mMiP, 2/8 were excluded because of anaemia at first ANC, and 1/8 had no Hb typing available). Those with subsequent mMiP were included until censoring in the survival analysis.

[**]Analysis stratified by literacy because of violation of proportional hazards for this variable.

[†]5/10 women in this group were excluded because they were anaemic at first ANC. In the whole group with severe Hb variants, 89.5 (95% CI 55.3–98.3) were anaemic at some point in pregnancy. Moderate and severe groups were combined for multivariable cox regression because of sparse data.

Abbreviations: ANC, antenatal care; HCT, haematocrit; mMiP, microscopically detected malaria in pregnancy; *P Plasmodium*; uPCR, ultrasensitive quantitative polymerase chain reaction.

birth weight in pregnancies with sMiP was 275 g (95% CI 40–510, *p* = 0.022) lower than that of pregnancies without malaria. For comparison, mean birth weight (corrected for gestational age and infant sex) in pregnancies with mMiP was 117 g (95% 47–188, *p* = 0.001) less than non-cases.

This association between sMiP and lower birth weight for a given gestational age and sex was seen for all species tested on univariable analysis (Tables 4 and S4). The association with lower birth weight remained after adjusting for confounders for sMiP overall (adjusted predicted mean difference −225 g, 95% CI −432 to −18, *p* = 0.033). Adjusted analysis by species showed a significant association with *P. falciparum* (adjusted predicted mean difference −154 g, 95% CI −235 to −72, *p* < 0.001) and unspeciated malaria parasitaemia (−606 g, 95% CI −1,128 to −84, *p* = 0.023), while the association with *P. vivax* (−77 g, 95% CI −163–10, *p* = 0.082) was weakened (Tables 4 and S4). The effect size of *P. falciparum* or sMiP overall was comparable to the effects of low maternal BMI (−86 g, 95% CI −136 to −37, *p* = 0.001) or smoking (−155 g, 95% CI −209 to −101, *p* < 0.001) in this cohort. The estimated effect size for unspeciated malaria parasitaemia was influenced strongly by one baby born at 1,070 g at

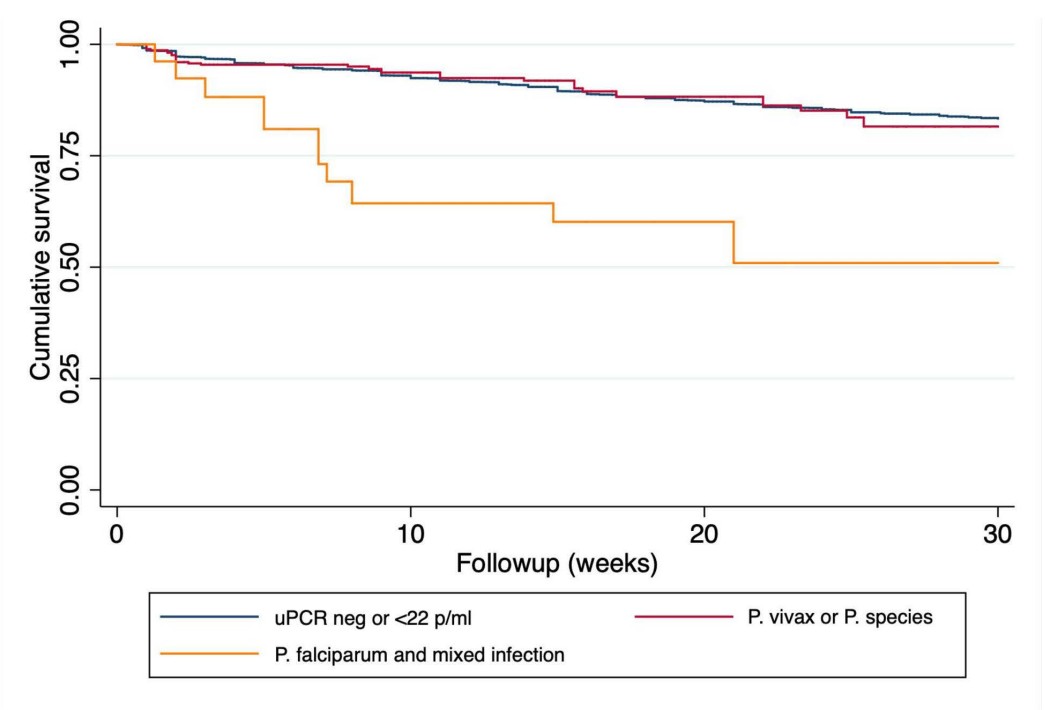

| | Follow-up interval (months) | | | | | | | |
|---|---|---|---|---|---|---|---|---|
| | 0-<1 | 1-<2 | 2-<3 | 3-<4 | 4-<5 | 5-<6 | 6-<7 | 7-<7.5 |
| uPCR neg at risk | 3668 | 2611 | 2193 | 1799 | 1405 | 1005 | 639 | 248 |
| uPCR neg anaemia | 281 | 67 | 79 | 60 | 44 | 33 | 25 | 10 |
| uPCR PV or P. sp at risk | 194 | 121 | 95 | 73 | 55 | 41 | 26 | 6 |
| uPCR PV or P. sp anaemia | 18 | 1 | 4 | 3 | 1 | 2 | 2 | 0 |
| uPCR PF or mix at risk | 27 | 14 | 8 | 7 | 6 | 4 | 2 | 2 |
| uPCR PF or mix anaemia | 8 | 4 | 1 | 1 | 0 | 1 | 0 | 0 |

**Fig 4. Weighted anaemia-free survival following a positive uPCR result by uPCR species detected.** Survival terminated at 30 weeks of follow up because of sparse data beyond that time.

35 weeks' gestational age in a pregnancy affected by pre-eclampsia. Exclusion of this outlier reduced the effect size, and the result was no longer statistically significant (adjusted HR −178 g, 95% CI −468–112, $p = 0.230$).

A sensitivity analysis removing the requirement of a dating ultrasound before 24 weeks (thereby increasing the total number of eligible pregnancies to 2,711) yielded a similar effect size for sMiP overall (adjusted predicted mean difference −139 g, 95% CI −284 to 6, $p = 0.060$), for *P. falciparum* (−153 g, 95% CI −236 to −70, $p < 0.001$), and for unspeciated malaria parasitaemia (−409 g, 95% CI −793 to −24, $p = 0.037$).

Numbers available for the analysis of birth weight were small, so subgroup analysis was done on two uPCR parasitaemia load groups, divided at the median value. Both lower and higher submicroscopic parasitaemia were associated with decreased birth weight compared with uninfected pregnancies without an apparent "dose-response" relationship, but the birth weight reduction was only statistically significant for the higher parasitaemia group: ≥368 p/

**Table 4. Association between submicroscopic malaria species at first antenatal care visit and birth weight.**

| | | Weighted mean birth weight in g (95% CI)* | Unadjusted mean predicted difference in birth weight (95% CI)† | p-value | Adjusted mean predicted difference in birth weight (95% CI)†† | p_value |
|---|---|---|---|---|---|---|
| **uPCR category** | | | | | | |
| Negative | | 3,081 (3,053–3,108) | Reference | – | Reference | – |
| *P. vivax* | | 2,926 (2,797–3,056) | *−141 (−246–37)* | *0.008* | −77 (−163–10) | 0.082 |
| *P. species (not differentiable)* | | 3,019 (2,778–3,260) | *−633 (−1,192 to −74)* | *0.020* | −606 (−118 to −84) | 0.023 |
| *P. falciparum* | | 2,899 (2,793–3,006) | *−141 (−232 to −50)* | *0.002* | −153 (−236 to −70) | <0.001 |
| **Status** | Refugee | 3,120 (3,088–3,151) | Reference | – | Reference | – |
| | Migrant | 3,037 (2,994–3,080) | −91 (−137 to −45) | 0.001 | −80 (−118 to −41) | <0.001 |
| **Gravidity** | Multigravida | 3,127 (3,094–3,160) | Reference | – | Reference | – |
| | Primigravida | 2,946 (2,902–2,989) | −157 (−207 to −106) | <0.001 | −185 (−228 to −141) | <0.001 |
| **BMI** | <18.5 | 2,903 (2,808–3,017) | −146 (−223 to −69) | <0.001 | −86 (−136 to −37) | 0.001 |
| | ≥ 18.5 | 3,103 (3,078–3,127) | Reference | – | Reference | – |
| **Year of enrolment** | 2012–2013 | 3,082 (3,021–3,142) | Reference | – | Reference | – |
| | 2014–2015 | 3,075 (3,050–3,100) | 16 (−40–72) | 0.574 | −8 (−52–36) | – |
| **Smoking** | Non–smoker | 3,092 (3,062–3,121) | Reference | – | Reference | – |
| | Smoker | 2,957 (2,892–3,021) | −129 (−184 to−73) | <0.001 | −155 (−209 to −101) | <0.001 |
| **Literacy** | Literate | 3,108 (3,080–3,137) | Reference | – | Reference | – |
| | Illiterate | 3,031 (2,979–3,083) | −62 (−111 to −12) | 0.015 | −38 (−80–4) | 0.074 |
| **Fetal number** | Singleton | 2,090 (3,066–3,113) | Reference | – | Reference | – |
| | Twin | 2,261 (2,161–2,361) | −491 (−628 to −353) | <0.001 | −452 (−591 to −313) | <0.001 |
| **Pre-eclampsia or eclampsia** | Absent | 3,081 (3,054–3,109) | Reference | – | Reference | – |
| | Present | 2,920 (2,781–3,058) | −147 (−417–124) | 0.288 | −61 (−220–98) | 0.451 |
| **Anaemia** | No anaemia | 3,084 (3,058–3,110) | Reference | – | – | – |
| | Anaemia | 3,047 (2,958–3,136) | −15 (−89–59) | 0.696 | – | – |

*N* = 1,940. Exclusions: any woman with microscopic malaria in pregnancy, congenital abnormality, stillbirth. Abbreviations: BMI, body mass index; uPCR, ultrasensitive quantitative polymerase chain reaction.

*Term (≥ 37 weeks + 0 days gestational age) only included in presentation of mean weights.

†Adjusted for sex and gestational age.

††Adjusted for sex, gestational age, status, gravidity, BMI, year of enrolment, Smoking, literacy, fetal number, pre-eclampsia, or eclampsia.

There was only one eligible neonate exposed to mixed submicroscopic infection with both *P. falciparum* and *P. vivax* in utero, so this neonate was excluded from the analysis.

mL (−134 g, 95% CI −264 to −4, *p* = 0.044) and 22–367 p/mL (adjusted predicted mean difference −270 g, 95% CI −562 to −21, *p* = 0.069).

There was no evidence of association between sMiP and preterm birth (S5 Table and S5B Fig).

The incidences of other rare adverse pregnancy outcomes, such as stillbirth (*n* = 37 in total, 1 with sMiP *P. vivax*), early neonatal death (*n* = 35 in total, none with sMiP), and maternal death (*n* = 8 in total, 1 with sMiP *P. vivax*), were too small to analyse in this cohort.

## Discussion

The adverse effects in pregnancy of microscopically detectable (i.e., parasite density > 50,000 p/mL) malaria have been extensively studied, but the effects of lower-density parasitaemia on pregnancy outcomes are less well characterised. This large study conducted in an area of low seasonal transmission shows conclusively that submicroscopic malaria parasitaemia in pregnancy is substantially more prevalent than microscopically detected parasitaemia, and

is associated with significant risks for the pregnancy. From 2012 to 2015, approximately 1 in 22 women on the Thailand–Myanmar border had asymptomatic sMiP at their first antenatal visit. In comparison, 1 in 100 had microscopically detected malaria at their first visit. Submicroscopic malaria parasitaemia was associated with an increased risk of subsequent microscopically detected malaria, anaemia, and decreased birth weight. There was no association with preterm birth, but otherwise the adverse pregnancy events and outcomes associated with this previously undetected much more prevalent form of malaria infection in pregnancy were similar to those documented previously with conventional malaria detection methods. Importantly, even parasite densities below those detectable with conventional capillary blood spot PCR (1,000–10,000 p/mL) were associated with these adverse outcomes.

The mean birth weight of infants of mothers who had sMiP at first ANC was more than 200 g below mean birth weight of non-case infants. Low birth weight is a risk factor for infant death. Mortality in the first 2 months of life is doubled for babies born small for gestational age [30]. Recent analyses have also found subsequent decreased school performance and reduced IQ at 5 years of age with decreasing birth weight [31,32]. This highlights the intergenerational effects of malaria, which persist despite intermittent preventive treatment in sub-Saharan Africa [33] or frequent testing and prompt treatment in this area of low seasonal transmission [1].

Although submicroscopic infection at first ANC visit increased the risk of developing subsequent mMiP 10-fold, most of the women with sMiP never manifested mMiP despite regular surveillance. Similar findings have been reported from Malawi where *P. falciparum* predominates [17] and Brazil where *P. vivax* predominates [13]. Both parasites may persist for months at low-fluctuating blood densities [34]. In *P. falciparum* malaria, the oscillations in single clone infections result from antigenic variation, whereas in the relapsing parasite *P. vivax* this is less well characterised. Why parasite densities should rise during pregnancy is unclear—the increasing natural immunosuppression of pregnancy is a likely contributor, although newly acquired infections and relapses of *P. vivax* may also contribute. Women with sMiP are likely to have a higher overall exposure to infectious bites [35]. Peripheral parasitaemia is known to be a poor reflection of placental parasitaemia in falciparum malaria, as parasites sequestered in the intervillous spaces of the placenta can evade detection by peripheral microscopy [36,37]. However, *P. vivax* does not sequester substantially and the mechanism of growth restriction caused by *P. vivax* is not well understood. The adverse outcomes associated with submicroscopic *P. falciparum* were generally greater than those for *P. vivax* in this study which may suggest that decreased birth weight resulting from patent *P. vivax* infection shown previously is largely mediated by clinically measurable factors such as fever, illness, or anaemia.

Submicroscopic *P. falciparum* (mono- or mixed) infection, in the absence of mMiP, was associated with increased risk of anaemia in this cohort in a low-transmission setting where regular follow-up and repeated screening for anaemia was available. This supports the findings of previous studies in other settings with low *P. falciparum* transmission [15,16]. The relationship between the severity of malaria infection (symptoms, recurrence, or degree of parasitaemia) and the risk of anaemia is well established [38]. Failure to identify sMiP in previous studies may have contributed to underestimation of the impact of malaria on anaemia in pregnancy. Although limited by small numbers, associations between mixed infections with *P. falciparum* and *P. vivax* and anaemia appeared to be similar in magnitude to associations between *P. falciparum* mono-infection and anaemia.

This study has limitations. This report describes the largest cohort of pregnant women with sMiP detected by high volume uPCR, but it is still underpowered. The cost of uPCR necessitated selection of samples for analysis and a case-cohort design. Late antenatal care attendance by some women limited the precision of the analysis of birth weight, as early ultrasound gives

the most accurate estimation of gestational age. Although efforts were made to elicit history of malaria treatment prior to first ANC, and antimalarials have been increasingly regulated in Thailand, self-treatment with antimalarial drugs cannot be entirely excluded. The setting of this study is both a strength and a weakness: the close follow-up, repeated screening, and early treatment of microscopically detected malaria episodes may lead to an underestimation of the impact of sMiP at first ANC visit on pregnancy outcomes. This level of intense follow-up is unusual in malaria endemic areas, where low-density infections may go untreated for longer periods of time. On the other hand, this intensive screening allowed this analysis to hone in on the specific impact of sMiP. The age of the cohort could be perceived to limit its relevance. However, the incidence of malaria in pregnancy at SMRU clinics is now more similar to 2012–2015 than it was in the intervening years, due to the increases in malaria transmission since the *coup d'etat* in Myanmar in 2021. This resurgence brings both microscopic and submicroscopic infections in pregnancy back to the top of the maternal and child health agenda. The implications of this study can also be relevant for other areas with similar malaria transmission.

Submicroscopic malaria infection is associated with both maternal and fetal ill health in malaria endemic areas, but what can be done now to detect it or prevent it? Replacing microscopy or rapid diagnostic tests (RDT) with uPCR cannot currently be offered or afforded in most settings. At a current price of 30 USD per test, screening all women at the SMRU clinics would cost over 100,000 USD per year at this small study site. The ultimate solution of malaria elimination is needed, but is still on the far horizon in many settings. In areas of higher transmission, a higher proportion of infections are detectable by RDT or microscopy [35], and prevention of infective bites and chemoprevention with effective and safe drugs reduces the negative impact of both high- and low-density infections. However, in areas with low and unstable transmission, or in areas targeted for malaria elimination, the relative contribution of sMiP to adverse outcomes is greater [6,35], as in this cohort. In the Greater Mekong subregion, where exophilic early evening or morning anopheline biting patterns prevail, the contribution of vector control to malaria prevention is less. The optimum strategy to prevent sMiP, other than malaria elimination in the general population, is still uncertain. In low transmission settings, determining this strategy requires consideration of the relative cost-effectiveness of chemoprevention, frequent antenatal clinic monitoring with conventional detection methods, targeted screening, or development and wider deployment of low-cost uPCR methods.

## Conclusions

Malaria in pregnancy is associated with adverse pregnancy outcomes in areas of low or unstable transmission, even when parasite densities are below the level of detection by conventional testing. Only malaria elimination can prevent definitively the multigenerational effects of malaria in pregnancy.

## Supporting information

**S1 Text. Details about data extraction and covariates are given.**
(DOCX)

**S2 Text. Details of the whole cohort of patients receiving antenatal care are given.**
(DOCX)

**S1 Fig. Weighted prevalence of submicroscopic malaria parasitaemia by gravidity with 95% CI.**
(TIF)

**S2 Fig. Submicroscopic malaria in pregnancy (sMiP) over the 4 years from 2012 to 2015.** There was a substantial decrease in the weighted proportion of sMiP for all species from the 4th quarter 2012 to the 4th quarter of 2015. (Only the 4th quarter—October–December—was sampled in all 4 years.). Abbreviations: *P.* species, *Plasmodium* species (not differentiable). (TIF)

**S3 Fig. Seasonality of submicroscopic malaria in pregnancy (sMiP) for six consecutive quarters from 4th quarter of 2013–1st quarter of 2015.** Weighted proportions of sMiP of each species were plotted against rainfall for six consecutive quarters from 4th quarter 2013–1st quarter 2015 to elucidate seasonality of sMiP. Abbreviations: *P.* species, *Plasmodium* species (not differentiable); sMiP submicroscopic malaria in pregnancy. Quarters: 1st Jan–Mar; 2nd Apr–Jun; 3rd Jul–Sep; 4th Oct–Dec. (TIF)

**S4 Fig. Weighted survival without anaemia following a positive ultrasensitive quantitative polymerase chain reaction (uPCR) result by uPCR species detected.** Here results are presented including mixed infection with *Plasmodium falciparum* and *vivax.* Survival terminated at 30 weeks of follow up because of sparse data beyond that time. (TIF)

**S5 Fig. Box and whisker plot of weighted mean birth weight (A) and gestational age (B) of women with accurate gestational age determined by ultrasound, grouped by level of malaria parasitaemia.** Abbreviations: uPCR, ultrasensitive quantitative polymerase chain reaction, neg, negative, sMiP, submicroscopic malaria in pregnancy, mMiP, microscopic malaria in pregnancy. Only term infants (gestational age ≥37 weeks + 0 days) were included in the birth weight figure. (TIF)

**S1 Table. Characteristics of women whose samples were tested for malaria by uPCR and women who were not included, after exclusion of cases with mMiP at first ANC.** (DOCX)

**S2 Table. Associations between baseline characteristics and submicroscopic malaria at first ANC.** (DOCX)

**S3 Table. Comparison of uPCR results at first ANC among women with a subsequent positive malaria smear.** The microscopic species was not always the same as the antecedent uPCR malaria species at first ANC visit. Nine out of the 10 episodes of mMiP following uPCR result of *P.* species, were *P. vivax.* Median (range) time from uPCR sample to microscopically detected malaria differed by uPCR result and submicroscopic species: 77 (7–232) days for uPCR negative ($n = 115$), 43 (7–203) days for *P. vivax* ($n = 49$), 32 (2–223) days for *P.* sp. ($n = 8$), 49 (42–51) days for *P. falciparum* ($n = 3$), and 49 (20–105) days for mixed infections ($n = 5$). (DOCX)

**S4 Table. Association between submicroscopic malaria species at first antenatal care visit and birth weight z-score.** (DOCX)

**S5 Table. Association between submicroscopic malaria species and gestational age at birth (excluding women with microscopic malaria in pregnancy (mMiP)) Because of the small numbers of events in eligible records (six preterm birth (PTB) for *P. falciparum*, *P.* species,**

**and mixed combined), survival analysis used submicroscopic infection with any species as the exposure.**
(DOCX)

**S1 Checklist.  STROBE checklist** .
(DOCX)

## Acknowledgments

The authors are indebted to the many laboratory, clinical, and IT staff at SMRU and MORU who were directly involved in creating and preserving the data presented here. In addition, many contributions of the logistics, administrative, and human resources departments support a system through which high-quality safe care can be given and high-quality data can be gathered. Above all, we thank the communities that accept and trust the services and studies conducted by SMRU and the families who trust us with their care at this critical stage of the life course. Finally, we are indebted to Sue Lee for her statistical guidance.

## Author contributions

**Conceptualisation:** Clare L. Ling, Nicholas J. White, François Nosten, Rose McGready.

**Data curation:** Mary Ellen Gilder, Warat Haohankhunnatham, Clare L. Ling, Gornpan Gornsawun, Germana Bancone, Peter R. Christensen, Nay Win Tun, Aung Myat Min, Verena I. Carrara, Stephane Proux, Rose McGready.

**Formal analysis:** Mary Ellen Gilder, Makoto Saito, Rose McGready.

**Funding acquisition:** Nicholas J. White, François Nosten, Rose McGready.

**Investigation:** Warat Haohankhunnatham, Clare L. Ling, Gornpan Gornsawun, Germana Bancone, Peter R. Christensen, Nay Win Tun, Aung Myat Min, Verena I. Carrara, Stephane Proux, Rose McGready.

**Methodology:** Mary Ellen Gilder, Makoto Saito, Warat Haohankhunnatham, Clare L. Ling, Gornpan Gornsawun, Germana Bancone, Mallika Imwong, Nicholas J. White, François Nosten, Rose McGready.

**Project administration:** Clare L. Ling, Nay Win Tun, Aung Myat Min, François Nosten, Rose McGready.

**Resources:** Prakaykaew Charunwatthana, François Nosten.

**Supervision:** Makoto Saito, Clare L. Ling, Germana Bancone, Cindy S. Chu, Prakaykaew Charunwatthana, Stephane Proux, Nicholas J. White, François Nosten, Rose McGready.

**Validation:** Mary Ellen Gilder, Warat Haohankhunnatham, Germana Bancone, Mallika Imwong, Stephane Proux.

**Visualisation:** Mary Ellen Gilder.

**Writing – original draft:** Mary Ellen Gilder.

**Writing – review & editing:** Mary Ellen Gilder, Makoto Saito, Clare L. Ling, Germana Bancone, Cindy S. Chu, Verena I. Carrara, Nicholas J. White, François Nosten, Rose McGready.

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
