## [Editor Report · Decision Letter 0]

15 Jul 2024

Dear Dr Gilder, 

Thank you for submitting your manuscript entitled "Adverse effects of submicroscopic malaria in pregnancy: a cohort study of 4,352 women on the Thailand-Myanmar border." for consideration by PLOS Medicine.

Your manuscript has now been evaluated by the PLOS Medicine editorial staff and I am writing to let you know that we would like to send your submission out for external peer review.

Please re-submit your manuscript within two working days, i.e. by Jul 17 2024 11:59PM.

Feel free to email me at lgaynor@plos.org if you have any queries relating to your submission.

Kind regards,

Louise Gaynor-Brook, MBBS PhD

Senior Editor

PLOS Medicine

---

## [Decision Letter · Decision Letter 1]

27 Aug 2024

Dear Dr Gilder,

Many thanks for submitting your manuscript "Adverse effects of submicroscopic malaria in pregnancy: a cohort study of 4,352 women on the Thailand-Myanmar border." (PMEDICINE-D-24-02236R1) to PLOS Medicine. The paper has been reviewed by subject experts and a statistician; their comments are included below and can also be accessed here: [LINK]

After discussing the paper with the editorial team and an academic editor with relevant expertise, I'm pleased to invite you to revise the paper in response to the reviewers' comments. We plan to send the revised paper to some or all of the original reviewers, and we cannot provide any guarantees at this stage regarding publication.

We ask that you submit your revision by Sep 17 2024 11:59PM. However, if this deadline is not feasible, please contact me by email, and we can discuss a suitable alternative.

Don't hesitate to contact me directly with any questions (lgaynor@plos.org). 

Best regards, 

Louise 

Louise Gaynor-Brook, MBBS PhD 

Senior Editor

PLOS Medicine

lgaynor@plos.org

Comments from the reviewers: 

Reviewer #1: This is a well written manuscript, with minimal grammatic/typo errors. In a case-cohort study, the report elegantly addresses the question if submicroscopic malaria affects inter-generational health outcomes during pregnancy.

Some points to clarify:

1) There are three groups derived from the case-cohort study, the last group regarded as "Control Group". Since the groups are derived based on outcomes, is it appropriate to designate the last group as "Control Group"? Why not just refer to it as "Group 3"?

2) For adjusting temporal changes in disease/infection prevalence, 8-time blocks were identified (lines 168-171). But the time blocks have not been characterized to show temporal trends, e.g., temporal trends of prevalence or parasitaemia. In lines 336-37, there is mention of primigravid women enrolled during early years of the case-cohort being at higher risk of getting a submicroscopic infection at their first ANC visit. This observation was not discussed considering the temporal variations.

3) Given that the uPCR has yielded parasitaemia ranges from 22 to 186,048p/ml, is there a high parasitaemic threshold above which parasitaemia is no more sMiP? In other words, is there uPCR threshold that precludes sMiP? In relation to this, can risk of developing mMiP be differentiated by parasitaemic groups: Q1, Q2, Q3, Q4? In line 250, the risk of developing mMiP, anemia and pre-term birth is given, but qualitatively based on presence or absence of sMiP. 

4) In line 277: the given date and numbers that follow can confuse the reader. Re-writing using a combination of numbers and letters/words can help. 

5) In lines 298-305, prevalence is given for the 3 groups: 36.1%, 5.1% and 3.8%. What was the mean parasitaemia for the 3 groups?

6) Given that the prevalence of mixed Pf/Pv malaria infection was only 1.8%, the analysis of data on risks (e.g., risk of anemia, lines 122/123) seems overly over-interpreted. This also seen in the tables showing HR and p-values.

7) In lines 132-135 (page 31), the fact that women were closely followed, screened for malaria and treated is discussed as a limitation of the study. It seems it should rather be a strength. Perhaps changing the context will help.

8) In table 3, Hb variants are classified as: Normal/mild, Moderate and Severe. This seems confusing as often Hb variants are mentioned in relation to abnormal structure of globulins and heme complex (Hg S, C, D, E). Maybe it is means to describe Hgb levels? 

Reviewer #2: The authors present a large study examining submicroscopic malaria in pregnancy and the risk of microscopic infection, birthweight and preterm birth. They have used a pragmatic approach of a case-cohort design given the associated cost of PCR for all samples and utilised an appropriate Cox regression. 

Minor comments: 

Postmenstrual age - should be estimated gestational age.

Tables 1 is difficult to follow. Why are exclusions included in the table? This would be better suited to a flow diagram 

Table 1: Most data has been presented categorically including maternal age and BMI. Including a measure of central tendency (mean/median dependent on distribution of data) and spread for continuous data will provide the reader with a better understanding of the population. 

Table 1: Were all data complete? If not, include number of missing for each variable 

Table 1: Given the case-cohort is the population and not whole cohort, I suggest moving the column of the whole cohort (11,901) to the supplemental, this will improve the readability of the table. I understand it was included to show random sampling was effective but can be included as a supplement. 

Table 1: It is unclear why the authors have presented weighted values for maternal demographics when they have also presented adjusted analyses to account for the differences. Additionally, weighting does not appear to change proportions substantially. 

Birthweight; the authors state in the methods intergrowth and z-scores were used but this does not appear to be so in the results. Adjusted analyses should consider including a measure of SES. The 95% CI for birthweight at term is very narrow, is this correct? Would expect a larger spread. 

Was haematocrit nadir normally distributed? 

Major comments

A significant limitation of the study is the age of the cohort, with samples collected 2012 - 2015 (ie up to 12 years old). Given the change in the population since this time, the relevance of the findings to current practice and population needs to be clarified. 

Throughout the authors have used IPW models but have not provided any details of how the models were constructed. How was balance assessed and what difference was considered acceptable? 

The authors have included selected confounders based on a p-value of <0.2. Why was this value selected? More importantly, including confounders purely based on p-values is not recommended, this can introduce bias. The use of an IPW model should consider what variables would predict being in the exposure group as well as consider confounders - not based on p-values but the use of subject knowledge and ideally, direct acyclic graphs. Were new models constructed for each maternal baseline characteristic? It is unclear which variables are included for each outcome. 

Reviewer #3: Overall this is a very interesting publication. 

minor comments- throughout the paper, all instances of microscopy detected malaria should be replaced by microscopically detected.

Page 38, "From 2012 to 2015, approximately 1 in 22 women attending their first antenatal visit on the Thailand-Myanmar border had asymptomatic sMiP at their first antenatal care visits." - delete one of the "first antenatal care visit"

Did you assess preterm delivery as a potential cause of some LBW?

I'm not entirely clear why you excluded women negative at ANC1? this could be better explained in the methods.

Page 41- this needs a ref: "In areas of higher transmission, a higher proportion of infections are detectable by RDT or microscopy"

* Please upload any figures associated with your paper as individual TIF or EPS files with 300dpi resolution at resubmission; please read our figure guidelines for more information on our requirements: http://journals.plos.org/plosmedicine/s/figures. While revising your submission, please upload your figure files to the PACE digital diagnostic tool, https://pacev2.apexcovantage.com/. PACE helps ensure that figures meet PLOS requirements. To use PACE, you must first register as a user. Then, login and navigate to the UPLOAD tab, where you will find detailed instructions on how to use the tool. If you encounter any issues or have any questions when using PACE, please email us at PLOSMedicine@plos.org.

FIGURES AND TABLES

SUPPLEMENTARY MATERIAL

REFERENCES

OBSERVATIONAL STUDIES

* Abstract: Please include the study design, population and setting, number of participants, years during which the study took place (enrollment and follow up), length of follow up, and main outcome measures.

* Please ensure that the study is reported according to the STROBE (or appropriate STROBE extension) guideline (available from: https://www.equator-network.org/reporting-guidelines/strobe) and include the completed STROBE (or STROBE extension) checklist as Supporting Information. Please add the following statement, or similar, to the Methods: "This study is reported as per the Strengthening the Reporting of Observational Studies in Epidemiology (STROBE) guideline (S1 Checklist)." When completing the checklist, please use section and paragraph numbers, rather than page numbers. 

* For all observational studies, in the manuscript text, please indicate: (1) the specific hypotheses you intended to test, (2) the analytical methods by which you planned to test them, (3) the analyses you actually performed, and (4) when reported analyses differ from those that were planned, transparent explanations for differences that affect the reliability of the study's results. If a reported analysis was performed based on an interesting but unanticipated pattern in the data, please be clear that the analysis was data driven. 

* Please state in the Methods section whether the study had a prospective protocol or analysis plan. If a prospective analysis plan (from your funding proposal, IRB or other ethics committee submission, study protocol, or other planning document written before analyzing the data) was used in designing the study, please include the relevant document(s) with your revised manuscript as a Supporting Information file to be published alongside your study and cite it in the Methods section. A legend for this file should be included at the end of your manuscript. If no such document exists, please make sure that the Methods section transparently describes when analyses were planned, and when/why any data-driven changes to analyses took place. Changes in the analysis, including those made in response to peer review comments, should be identified as such in the Methods section of the paper, with rationale.

---

## [Decision Letter · Decision Letter 2]

4 Oct 2024

Dear Dr Gilder,

Many thanks for submitting your manuscript "Adverse effects of submicroscopic malaria in pregnancy: a cohort study of 4,352 women on the Thailand-Myanmar border." (PMEDICINE-D-24-02236R2) to PLOS Medicine. The paper has been reviewed by subject experts and a statistician; their comments are included below and can also be accessed here: [LINK]

As you will see, the statistical reviewer has requested further information regarding which variables have been included for each IPW model, and how covariate balance was checked between exposure groups. After discussing the paper with the editorial team, I'm pleased to invite you to revise the paper in response to the reviewer's comments. We plan to send the revised paper to some or all of the original reviewers, and we cannot provide any guarantees at this stage regarding publication.

We ask that you submit your revision by Oct 25 2024 11:59PM. However, if this deadline is not feasible, please contact me by email, and we can discuss a suitable alternative.

Don't hesitate to contact me directly with any questions (lgaynor@plos.org). 

Best regards, 

Louise 

Louise Gaynor-Brook, MBBS PhD 

Senior Editor

PLOS Medicine

lgaynor@plos.org

Comments from the reviewers: 

Reviewer #1: None

Reviewer #2: I thank the authors for their updates to the manuscript. 

About the adjusted analyses, it is still not clear exactly which variables have been included for each IPW model. Can the authors include their models as a supplement? Removing variables that are considered theoretical confounders due to a p-value in their dataset is not recommended, which variables were removed based on p-values? Given you are using an IPW model the number of variables included in the model is less of a concern than in traditional adjusted regression. For these IPW cox regression models, how was covariate balance checked between exposure groups and what level determined acceptable?

* Please upload any figures associated with your paper as individual TIF or EPS files with 300dpi resolution at resubmission; please read our figure guidelines for more information on our requirements: http://journals.plos.org/plosmedicine/s/figures. While revising your submission, please upload your figure files to the PACE digital diagnostic tool, https://pacev2.apexcovantage.com/. PACE helps ensure that figures meet PLOS requirements. To use PACE, you must first register as a user. Then, login and navigate to the UPLOAD tab, where you will find detailed instructions on how to use the tool. If you encounter any issues or have any questions when using PACE, please email us at PLOSMedicine@plos.org.

FIGURES AND TABLES

SUPPLEMENTARY MATERIAL

REFERENCES

OBSERVATIONAL STUDIES

* Abstract: Please include the study design, population and setting, number of participants, years during which the study took place (enrollment and follow up), length of follow up, and main outcome measures.

* Please ensure that the study is reported according to the STROBE (or appropriate STOBE extension) guideline (available from: https://www.equator-network.org/reporting-guidelines/strobe) and include the completed STROBE (or STROBE extension) checklist as Supporting Information. Please add the following statement, or similar, to the Methods: "This study is reported as per the Strengthening the Reporting of Observational Studies in Epidemiology (STROBE) guideline (S1 Checklist)." When completing the checklist, please use section and paragraph numbers, rather than page numbers. 

* For all observational studies, in the manuscript text, please indicate: (1) the specific hypotheses you intended to test, (2) the analytical methods by which you planned to test them, (3) the analyses you actually performed, and (4) when reported analyses differ from those that were planned, transparent explanations for differences that affect the reliability of the study's results. If a reported analysis was performed based on an interesting but unanticipated pattern in the data, please be clear that the analysis was data driven. 

* Please state in the Methods section whether the study had a prospective protocol or analysis plan. If a prospective analysis plan (from your funding proposal, IRB or other ethics committee submission, study protocol, or other planning document written before analyzing the data) was used in designing the study, please include the relevant document(s) with your revised manuscript as a Supporting Information file to be published alongside your study and cite it in the Methods section. A legend for this file should be included at the end of your manuscript. If no such document exists, please make sure that the Methods section transparently describes when analyses were planned, and when/why any data-driven changes to analyses took place. Changes in the analysis, including those made in response to peer review comments, should be identified as such in the Methods section of the paper, with rationale.

---

## [Decision Letter · Decision Letter 3]

16 Dec 2024

Dear Dr. Gilder,

Thank you very much for re-submitting your manuscript "Adverse effects of submicroscopic malaria in pregnancy: a cohort study of 4,352 women on the Thailand-Myanmar border." (PMEDICINE-D-24-02236R3) for review by PLOS Medicine.

I have discussed the paper with my colleagues and the academic editor and it was also seen again by one reviewer. I am pleased to say that provided the remaining editorial and production issues are dealt with we are planning to accept the paper for publication in the journal.

[LINK]

We expect to receive your revised manuscript within the next few weeks. Please email us (plosmedicine@plos.org) if you have any questions or concerns.

We look forward to receiving the revised manuscript by Jan 02 2025 11:59PM.   

Sincerely,

Rebecca Kirk

On behalf of:

Louise Gaynor-Brook, MBBS PhD

Senior Editor 

PLOS Medicine

plosmedicine.org

Requests from Editors:

GENERAL EDITORIAL REQEUSTS

* At this stage, we ask that you include a short, non-technical Author Summary of your research to make findings accessible to a wide audience that includes both scientists and non-scientists. The Author Summary should immediately follow the Abstract in your revised manuscript. This text is subject to editorial change and should be distinct from the scientific abstract. Ideally each sub-heading should contain 2-3 single sentence, concise bullet points containing the most salient points from your study. In the final bullet point of ‘What Do These Findings Mean?’ Please include the main limitations of the study in non-technical language.

Please see our author guidelines for more information: https://journals.plos.org/plosmedicine/s/revising-your-manuscript#loc-author-summary.

* Please confirm that your title complies with to PLOS Medicine's style. Your title must be nondeclarative and not a question. It should begin with main concept if possible. "Effect of" should be used only if causality can be inferred, i.e., for an RCT. Please place the study design ("A randomized controlled trial," "A retrospective study," "A modelling study," etc.) in the subtitle (ie, after a colon).

* Please confirm that your abstract complies with our requirements, including providing all the information relevant to this study type https://journals.plos.org/plosmedicine/s/submission-guidelines#loc-abstract

* Please ensure that the Introduction ends with a clear description of the study question or hypothesis.

* Please ensure that all abbreviations are defined at first use throughout the text.

GENERAL

* Please review your text for claims of novelty or primacy and remove this language. In addition, please check that any use of statistical terms (such as trend or significant) are supported by the data, and if not please remove them.

FUNDING STATEMENT

* The funding statement should include: specific grant numbers, initials of authors who received each award, URLs to sponsors’ websites. Also, please state whether any sponsors or funders (other than the named authors) played any role in study design, data collection and analysis, the decision to publish, or preparation of the manuscript. If they had no role in the research, include this sentence: “The funders had no role in study design, data collection and analysis, decision to publish, or preparation of the manuscript.”

COMPETING INTERESTS STATEMENT

* All authors must declare their relevant competing interests per the PLOS policy, which can be seen here: https://journals.plos.org/plosmedicine/s/competing-interests For authors with ties to industry, please indicate whether any of the interests has a financial stake in the results of the current study.

OBSERVATIONAL, COHORT, CROSS-SECTIONAL, AND CASE CONTROL STUDIES

* Please ensure that the study is reported according to the STROBE guideline, and include the completed STROBE checklist as Supporting Information. Please add the following statement, or similar, to the Methods: ""This study is reported as per the Strengthening the Reporting of Observational Studies in Epidemiology (STROBE) guideline (S1 Checklist)."

When completing the checklist, please use section and paragraph numbers, rather than page numbers."

* Did your study have a prospective protocol or analysis plan? Please state this (either way) early in the Methods section.

* Your study is observational and therefore causality cannot be inferred. Please remove language that implies causality and refer to associations instead.

Comments from Reviewers:

Reviewer #2: I thank the authors for their reply and updated manuscript/regression following further thought given to potential confounders. Although I believe the adjustment is sufficient the DAGs have not been constructed correctly, with multiple variables included as one, variables adjusted for but not in DAG (such as sex), adjusted variables shown rather than the total model and interactions between variables not entirely considered. However, these updates are unlikely to change the adjusted models and I therefore suggest the DAGs are removed from the figures.

[LINK]

---

## [Editor Report · Decision Letter 4]

10 Jan 2025

Dear Dr Gilder, 

On behalf of my colleagues and the Academic Editor, James Beeson, I am pleased to inform you that we have agreed to publish your manuscript "Submicroscopic malaria in pregnancy and associated adverse pregnancy events: a case-cohort study of 4,352 women on the Thailand-Myanmar border." (PMEDICINE-D-24-02236R4) in PLOS Medicine.

PRESS

Sincerely, 

Rebecca Kirk

On behalf of:

Louise Gaynor-Brook, MBBS PhD 

Senior Editor 

PLOS Medicine